# Ignition of Deposited Wood Dust Layer by Selected Sources

**Ivana Tureková [1] and Iveta Marková [2,*]**

[1] Department of Technology and Information Technologies, Constantine the Philosopher University in Nitra, Tr. A. Hlinku 1, 949 74 Nitra, Slovakia; iturekova@ukf.sk

[2] Department of Fire Engineering, Faculty of Security Engineering, University of Žilina, Univerzitná 1, 010 26 Žilina, Slovakia

* Correspondence: iveta.markova@fbi.uniza.sk; Tel.: +421-041-513-6799

**Abstract:** The main waste of wood sanding technology is wood dust. The formation of wood dust affects its behaviour. Wood dust can be in a turbulent form and behaves explosively or in a settled form where it becomes flammable. Dust particles are barely detectable by the naked eye, wood dust still presents substantial health, safety, fire and explosion risks to employees. This article deals with the evaluation of ignition temperature and surface temperature of deposited wood dust samples by selected ignition sources. The influence of selected physical properties of wood dust, the size of the contact area between the ignition source and the combustible material, the spatial arrangement during the ignition and the application time of the ignition source are analysed. The paper describes the behaviour of a 15 mm deposited layer of wood dust of spruce (*Picea abies* L.), beech (*Fagus silvatica* L.). oak (*Quercus petraea* Liebl.) caused by three potential ignition sources—a hot surface, an electric coil and a smouldering cigarette. Prior to the experimental determination of the ignition temperature, dust moisture content which did not significantly affect the ignition phase of the samples, as well as sieve analysis of tested samples were determined. The lowest minimum ignition temperature on the hot plate, as an important property of any fuel, because the combustion reaction of the fuel becomes self-sustaining only above this temperature, was reached by the oak dust sample (280 °C), the highest by the spruce dust sample (300 °C). The ignition process of wood dust was comparable in all samples, differing in the ignition time and the area of the thermally degraded layer. The least effective ignition source was a smouldering cigarette.

**Keywords:** wood dust; minimum ignition temperature; hot plate; electric coil; smouldering cigarette

## 1. Introduction

Slovakia may be a small country but it has a large forest area, which represents 41% of the total country area (for forest land), i.e., about two million hectares (Forest is the natural heritage of our country) beech is the most common tree species naturally occurring in the Slovak Republic. It grows most often along with fir and spruce. Such composition of tree species is called "Carpathian mixture". Spruce represents 26.4% of the tree population in our country, while the coniferous trees are among the most common [1].

The woodworking industry is one of the sectors where dust is generated as unwanted waste. It gets into the environment mainly from the processes of shredding, cutting and grinding of wood [2,3]. Wood dust is classified as combustible organic dust [4–7]. Dustiness is the number of dust particles contained in the volume unit of gas (air). Dustiness means the tendency of a powder to form airborne dust by a prescribed mechanical stimulus [8,9]. The dustiness of the primary step occurs in manufacturing processes (such as wood dust). The next step of dustiness is generated by the swirling of settled dust

from primary dustiness. Dustiness is given by the number of particles (numerically) or the mass of particles (gravimetrically) in a given volume in µg·m$^{-3}$ [10–12]. Dustiness in the environment has harmful effects on the human organism [13,14] and represents one of the fundamental problems in the field of occupational safety and hygiene [15], not excluding the environment. Hygienic aspects of dust presented by the conventions of dust in terms of CEN Standard [16] and US-EPA present the character of dust according to its size. The fraction size of 20–30 µm comes into attention, where the possibility of inhalation of wood dust occurs [17].

The chemical composition of waste wood dust does not differ from the chemical composition of the solid wood from which the dust has been produced. Both the chemical structure and elemental composition of wood remain unchanged after its disintegration. The results of Vandličková et al. [18], Mračková [19], Marková et al. [20] and Mračková and Marková [21] point out that dust particles also retain the morphology of the wood structure.

In the field of fire safety, it is necessary to understand the behaviour of combustible materials under external heat flux [22]. There are three necessary preconditions for a fire to take place (combustible substance, oxidising agent and ignition source) and they are a basis for fire ignition and propagation [23]. The influence/effectiveness of the ignition source depends on the nature of the combustible set (mixture of combustible substance and oxidising agent—air) [24]. The existence of a combustible mixture, i.e., a suitable quantitative ratio of a combustible substance to oxygen, is a platform for the generation and development of combustion [25].

The ignition source must contain sufficient energy to ignite a combustible mixture [26]. After initiation of the combustion process, it is necessary to continue with sufficient heat input into the combustion zone (Demidov's theory in [27,28] or along with other heat sources [29,30]). Initiation of the combustion process is the most important step in this process. Bond [31] provides a detailed description of the ignition sources. According to the source of the ignition, there are three types of process initiations [32]:

- Spontaneous ignition due to an external source of radiant heat;
- Initiation by external source of ignition (open flame, spark);
- Spontaneous ignition without an effect of an external heat source (autoignition, chemical reaction).

The ignition and autoignition of solid materials may be considered a transient phenomenon, which depends on temperature, heating and self-heating conditions and heat accumulation. Conditions for initiating the combustion process are limited, particularly by the concentration of fuel and oxygen and by the source of ignition (flame, radiant heat, spark, etc.) [33,34].

Methods for determining the ability of materials to ignite are based on the determination of the boundary conditions at which the ignition or autoignition is observed.

The ignition source of a given combustible system may be an object or a substance which has a certain temperature and is capable of transmitting the necessary amount of energy of a specific type for a certain period of time [35–37]. The ignition energy is supplied to the combustible mixture by an external ignition source. Hence, it is called "external" ignition. There are substances (self-igniting) that are able to self-activate due to their individual instability [38] or chemical oxidation.

Dust mixtures are dispersion systems which exhibit similar properties to gas mixtures in the case of a higher degree of dispersion. The combustion or explosion of dust and dust mixtures is essentially governed by the same laws as gas mixtures. Deposited dust may burn in a form of flaming combustion, glowing or smouldering [39,40].

Combustible dust hazard may occur especially in places where dust deposits in a continuous layer which is capable of spreading the flame. Any combustible dust fire can very easily transition to an explosion and vice versa, an explosion of combustible dust can transition to the combustion of the remaining unreacted dust. However, the explosion need not be followed by fire if the explosion

consumes oxygen in the air, alternatively if the oxygen content in the given area is significantly reduced [41]. The ignition risk of a deposited layer of dust may arise if [42]:

$$t_{PRAC} \geq k_b t_{MIN}^u \qquad (1)$$

where $t_{PRAC}$—hot surface temperature in °C, $t^u{}_{MIN}$—minimum temperature of deposited dust in °C, $k_b$—a safety coefficient amounting 2/3 by the Damec [42].

Combustible dust in the whirling state is capable of having a violent oxidation reaction while this reaction has the character of an explosion and under certain conditions, this process can turn into a detonation [41]. The detonation takes place at a speed higher than 1000 m·s$^{-1}$ and at the head of the pressure field, there is a shock wave, immediate pressure increases to the value of units up to tens of GPa (type of explosion) [42].

The aim of this paper is to study the effect of ignition sources on wood species of spruce, beech and oak dust.

## 2. Materials and Methods

For the preparation of wood dust, 3 round wood pieces were prepared from each wood type (300 × 50 × 50 mm). The wood pieces were dried to a humidity of approx. 8–10% [43].

The ignition of wood dust from spruce, oak and beech which are the most commonly used wood species in the industry, is monitored depending on the ignition source (a hot surface, an electric coil, a cigarette), size of the contact area between ignition source and surface wood dust contact with the ignition source. The research follows up on solved Slovak research projects of wood dust (such as [1,20,44]).

### 2.1. Wood Dust Samples—Preparation Process

The waste samples were selected with regards to the most common wood types in the industrial processing in furniture plants and, at the same time, they represent a fair sample of individual tree species. Specifically, they include coniferous species—Spruce (*Picea abies* L.), deciduous ring-porous species—Winter Oak (*Quercus petraea* Liebl.) and deciduous diffuse-porous species—Beech (*Fagus silvatica* L.). The waste spruce samples (sawdust) were produced using a circular saw DMMA—36 (Reszelskie Zakłady Przemysłu Maszynowego Leśnictwa w Reszlu, Reszel, Poland). Dust used in the study was not waste. It was produced especially for this purpose. In the selection of samples, the emphasis was on samples that contain as few knots in the wood as possible and had the same increment of annual rings. Samples were tangentially sawn timber. The methods and forms of circular sawing will be described in detail Siklienka and Mišura [45].

Waste beech dust was produced by grinding with a rough grinding wheel (BOSCH), grain size B126. Waste oak wood dust samples (fine grind) were produced using a BOSCH pss 200 ac orbital sander (Robert Bosch GmbH, Gerlingen, Germany). The samples were prepared by an experienced grinding expert in order to bring the grinding process as close to reality as possible in terms of grinding agent pressure on the workpiece surface, grinding speed as well as grinding direction (cross). Abrasive paper with grain size P80 (Norton P80 H231) was used during the experiments. The difference in the particle composition of the prepared dust samples can be seen (Figure 1) in the performed sieve analysis. The bag was cleared after each wood dust species by a vacuum cleaner Kärcher WD 2 (Alfred Kärcher SE and Co. KG, Winnenden, Germany). Different types of wood dust were divided into nine fractions according to the sieve mesh size.

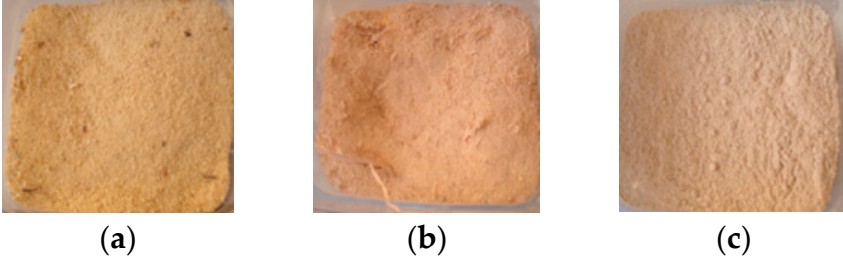

**Figure 1.** Waste wood dust samples. Legend: (**a**) spruce, (**b**) beech, (**c**) oak.

*2.2. Physical Properties of Wood Dusts and Sieve Analysis*

As the water content of waste wood dust (determined by a gravimetric method [46]) increases, the resistance of the dust to ignition also increases ([47,48]. Part of the supplied energy is used to evaporate free water, break bonds and subsequently evaporate bound and chemically bound water. Combustible gases diluted with water vapour have a lower concentration and thus lower flammability. The higher the moisture content of dust, the higher the ignition time [49].

Sieve analysis in accordance with [50] was carried out by an automatic vibratory sieve shaker Retsch AS 200 control and with a set of control stainless steel sieves: diameter of sieve, 200 mm; height, 50 mm; diameter of sieve mesh. <0.056; 0.056; 0.071; 0.090; 0.150; 0.200; 0.250; 0.500 (mm). Shares of residues on each sieve and bottom were researched using digital laboratory balance Radwag KERN PLT (KERN and SOHN, Balingen, Germany) with accuracy of weighing 0.001 g. Sieving parameters include: amplitude 2 mm·g$^{-1}$, with an interval of 10 s, sieving time 20 min. The measurement process was realised by [50].

*2.3. Determining the Minimum Ignition Temperature According to EN 50281 of Waste Wood Dust Layer Samples from Hot Surfaces*

The minimum ignition temperature was determined on an electrically hot plate (Figure 2) 185 mm in diameter according to EN 50281-2-1: 2002 Methods for determining the minimum ignition temperatures of dust [51]. This standard defines minimum temperature for ignition of a dust layer as a minimum temperature of a hot surface by which ignition of a dust layer can occur with a setting thickness located on that hot surface [47]. According to the procedure EN 50281-2-1:2002 (Method A), the ignition of dust in the layer could occur when [51]:

(a)    Glowing or flame combustion was observed;
(b)    Measured temperature of dust have reached 450 °C;
(c)    Measured temperature of dust exceeded 250 K (the temperature of the furnace plate).

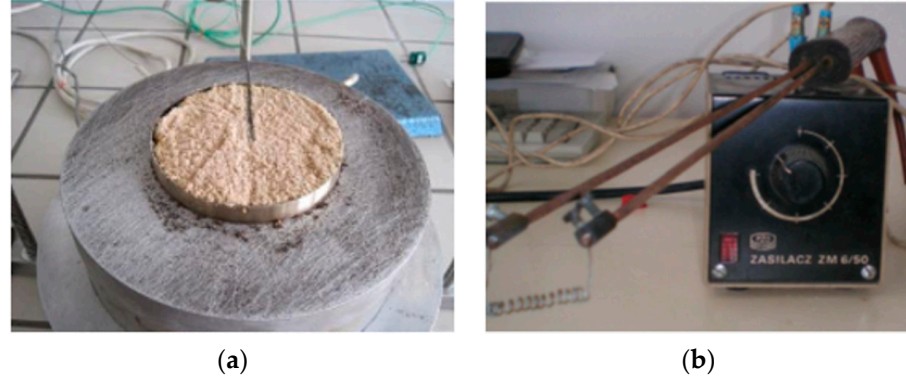

**Figure 2.** The test equipment establishing minimum ignition temperatures of the dust layer. Legend: (**a**) Hot plate apparatus with full metal ring of beech dust; (**b**) electric coil with a direct current source (TTAD3, JLT Slovakia).

In reference to point (b) and (c), the ignition of dust has not occurred if it could be indicated that oxidise reaction does not change into glowing or flame combustion process.

It characterises dust according to its ability to ignite in a layer under heat stress from a hot environment or by contact with another hot surface [47].

The hot plate assembly temperature is set and controlled to the desired temperature by a temperature controller. The temperature of the hot plate assembly and the dust layer sample are continuously monitored and recorded on a data logger/data sheet as a function of time to the end of the test. The test is continued until the sample layer changes into glowing or flame combustion process ignites, or reaches a maximum temperature without igniting, and is cooled down. Visual observations are also made during the test. If after 120 min, self-heating is not apparent, the test is terminated [47].

The height of the waste wood dust layer sample placed on the surface of the hot plate was 15 mm. The layer height was chosen to be the highest, which is probably in grinding processes. At the same time, the stated layer height is recommended in the methodology [47] and in the references [43,49]. The ambient temperature was 22.6 °C. The exposure time of the hot surface to the individual dust samples was individually set as described in detail in Table 1.

**Table 1.** Conditions of experiments carried out in the hot plate apparatus.

| Ignition Sources | Samples | Dust Layer (mm) | Contact Ignition Source/Dust | Temperature of Ignition Source (°C) | Time of Experiment |
|---|---|---|---|---|---|
| Hot surface | Spruce Beech Oak | 15 | Ring area d = 185 mm | <300 | 120 min |
| Electrical coil | Spruce Beech Oak | 15 | 2 height points (7.5 ± 1) mm over the bottom of the deposited layer | 490 | 180 s |
| Smouldering cigarette | Spruce Beech Oak | 15 | 1 point in the centre of the ring area | 300–450 | 120 s |

The minimum ignition temperature is defined as the lowest surface temperature of a hot plate at which at least one of the following phenomena can be observed during the test [51,52]:

- Glowing, smouldering or flaming combustion;
- The temperature–time curve obtained by the thermocouple placed in the centre of the dust layer increases steadily along with the temperature of the isothermal hot plate.

The sample temperature is monitored to determine temperature rise due to oxidative and/or decomposition reactions. According to Dastidar [47], ignition is considered to have taken place when either of the following occurs [47]:

- There is visible evidence of combustion such as a red glow or a flame;
- The temperature in the dust layer at the position of the thermocouple rises at least 50 °C above the hot plate temperature.

The other tested parameter was charred area (%). The parameter was stated for experiments with electric coil and smouldering cigarette thermal source. Thermal source was put to the centre of of the dust layer. The charred area has gradually expanded on the hot plate from influence on resources. The finished position of the charred area was documented by photo and the photo was basic for calculation of the charred area.

All experiments were performed on a laboratory hot plate apparatus with temperature controller type CLARE 4.0 (Clasic, Praguea, Czech Republic). This test method covers a laboratory procedure to determine the hot surface ignition temperature of dust layers [46] by measuring the minimum temperature at which a dust layer will self-heat. The data obtained from this test method provides a relative measure of the hot surface ignition temperature of a dust layer. The hot surface ignition temperature of the material in the form of a dust layer is determined by using a hot plate apparatus. In all cases, the dust consists of approximately 90% of particles finer than 0.5 mm ($d_{50}$). The heated surface (hot plate) consists of a 20 mm thick circular metal plate with a working surface of 200 mm. The metal plate is electrically heated and its temperature is regulated by a thermocouple located in the middle of the plate. A second thermocouple is placed parallel to the heated surface, just below the

surface of the dust layer. During the measurement, the plate temperature and the temperature just below the dust layer are recorded.

The experiments with waste wood dust ignition were carried out with different minimum ignition resources, namely a hot surface, an electric coil and a smouldering cigarette (Table 1).

### 2.4. Ignition of Waste Wood Dust Samples by an Electric Coil

Ignition (Figure 2a) with a DC power supply of 490 °C (measured with a Ni-Cr-Ni thermocouple) was performed on a hot plate laboratory apparatus (Model VmWare, VWR Scientific, Thorofore, NJ, USA). The coil ignition took 180 s. During the experiment, the temperature behaviour of the samples at the surface and at the bottom of the dust layer was monitored by two thermocouples (Measurement Point 1, Measurement Point 2; Figure 3). Two types of thermocouples and a data loggers (Model PHL22B12, Fuji Electric Systems Co., Ltd., Tokyo, Japan) were used to continuously monitor and record the temperature of the metal plate and the dust layer in the metal ring. The electric coil was covered with the sample and acted at a height of 7.5 ± 1 mm above the bottom of the deposited layer and at the same time above the measuring point No. 2 (Figure 3b).

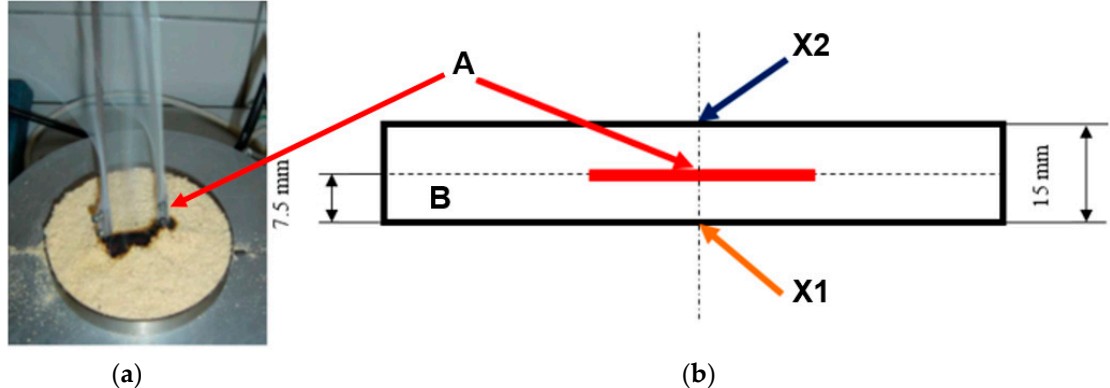

|   (**a**)   |   (**b**)   |

**Figure 3.** (**a**) Photo of inserting an electric coil into a dust layer; (**b**) Inserting an electric coil into the sample layer—diagram (side view). Legend: A—inserting an electric coil, B—dust layer, X1—measurement point of temperature hot plate, X2—measurement point of the temperature dust layer.

The electric coil initiated thermal degradation of wood dust (Figure 3). Thermocouples (Model PHL22B12, Fuji Electric Systems Co., Ltd., Tokyo, Japan) document the course of temperatures (Figure 3b) of wood dust on the surface of the hot plate (measurement point X1) and at a height (of 7.5 ± 1 mm) above the bottom of the deposited layer (measurement point X2).

Cigarettes are a common cause of fire [53,54]. An unextinguished cigarette butt was used as an ignition source in the experiment. It was placed on the surface of the waste wood dust so that the tip of the burning cigarette, i.e., the region with the highest temperature proceeds from the centre towards the edge of the layer. The research was carried out in the same manner as in the previous experiments. Each experiment was repeated 5 times.

### 3. Results and Discussion

### 3.1. Determination of Moisture Content of Waste Wood Dust

The spruce dust sample had value moisture higher than others (Table 2). The samples contained sawdust, which was related to the technological processing of wood by cutting. Sawdust and chips have a higher proportion of bound water and they are harder to dry than fine dust particles [52,55]. However, the moisture values which were determined should not have a significant effect on the results of wood dust ignition experiments [48,49,56]. The hygroscopic properties of wood dust in relation to its size were already studied [57].

**Table 2.** Granulometrically determined humidity of individual samples of waste wood dust.

| Moisture | Waste Wood Dust Samples | | |
|---|---|---|---|
| | **Spruce** | **Beech** | **Oak** |
| (%) | 6.428 ± 0.025 | 5.085 ± 0.012 | 4.793 ± 0.020 |

### 3.2. Sieve Analysis

The results of dust sieving are presented by cumulative curves (Figure 4). The results are expressed in weight percent of the individual fractions collected on a sieve with the appropriate mesh size. The results of the sieve analysis should be presented in accordance with the standard ISO 9276-1 [58] and [59]. Prepared Continuous Cumulative Curves are presented in Figure 4 with complete mesh size.

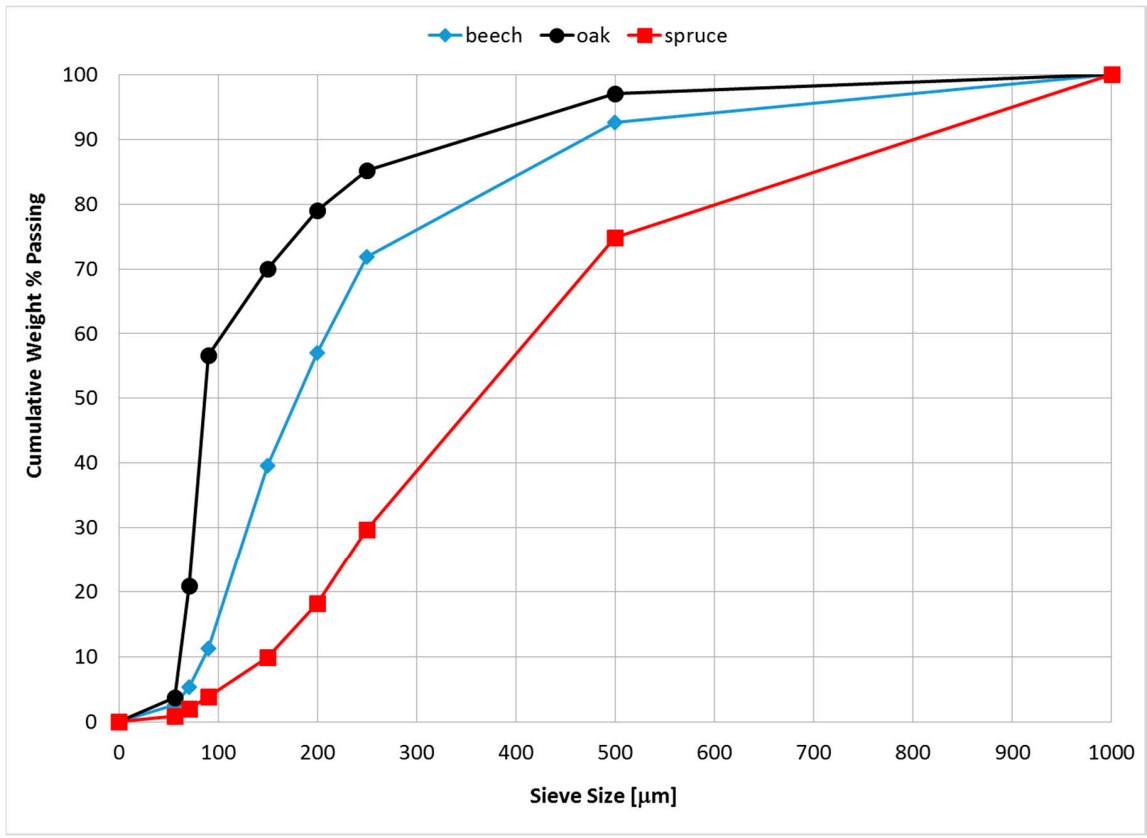

**Figure 4.** Continuous cumulative curve of spruce, beech and oak dust.

The granulometric composition of the individual samples is influenced by the nature of the abrasive processes by which the samples were produced and the type of trees which were used [1,20,55,56]. The examined samples of dust were produced in various technological operations, so their granulometric composition also differs.

Spruce dust, as a product of a circular saw cutting (BOSCH GKS 190, manufactured by BOSCH, Gerlingen, Germany), had the highest proportion of 250 μm to 500 μm (45%) fraction, containing also small and larger chips, which was also identified by a study of its microstructure (Figure 4). Sawing is an operation where larger particles are produced (Figure 4). Dzurenda and Orlowski [60] carried out an analysis of the shape, dimensions and granulometric composition of sawdust produced in the longitudinal sawing process of dry spruce timber by a thin cutting frame saw, type: CLASIC 150/200 (at material feed speed v = 0.5 m·min$^{-1}$). The results of the sieve analysis characterise dry spruce sawdust as a polydisperse bulk material with a grain size in the range of 85.38 μm to 28.2 μm,

and the largest fraction in the spruce sawdust with grain size in the range of d = 125 to 1000 μm, which makes up 86.77–87.15% of the extracted sawdust from the thin cutting frame saws. These results are also supported by Kučerka [61] in the study of dry spruce and oak sawdust produced in wood sawing processes.

Beech dust had the largest fraction of 90 μm to 150 μm (28%). The finest dust was produced from oak (Figure 4) with the largest fraction of 71 μm to 90 μm.

### 3.3. Ignition of Waste Wood Dust

### 3.3.1. Monitoring of Hot Surface Ignition

The set of experiments was preceded by indicative tests to determine the height of the waste wood dust layer on the hot plate apparatus. The first degradation processes were observed near the test ring (Figure 5), which was used to define the height of the dust layer. Our aim was to remove the effect of the heating from the side wall, because a body with high thermal conductivity which is in contact with a hot surface and reaches a layer of deposited dust can cause the dust to heat up. This effect could not be completely avoided, in all cases of heating by a hot plate apparatus, smouldering nests were formed first around the perimeter of the sample (Figure 5).

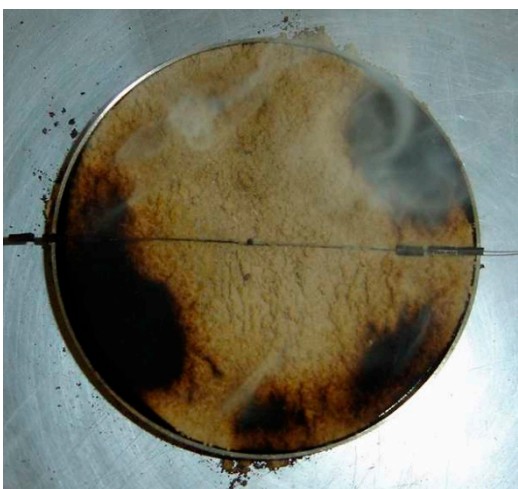

**Figure 5.** Smouldering nests around the perimeter of the sample caused by the ignition source—hot surface at the 10th min.

The ignition of a dust layer by a hot surface at a certain temperature depends strongly on the balance between the rate of heat generation in the layer and the rate of heat release to the environment [62,63]. The essence of the process results from a comparison of the surface temperature of the hot plate with the minimum ignition temperature of deposited dust (Figure 6). Obtaining sufficient heat from the hot surface to heat the deposited dust to a minimum ignition temperature depends on the amount of dust, i.e., the thickness of the deposited layer. The ignition temperature of a given material, therefore, depends on the thickness of the deposited layer [48]. The purpose of this research was to monitor the influence of selected ignition sources on the deposited layer of dust, and therefore, a uniform thickness of 15 mm was chosen.

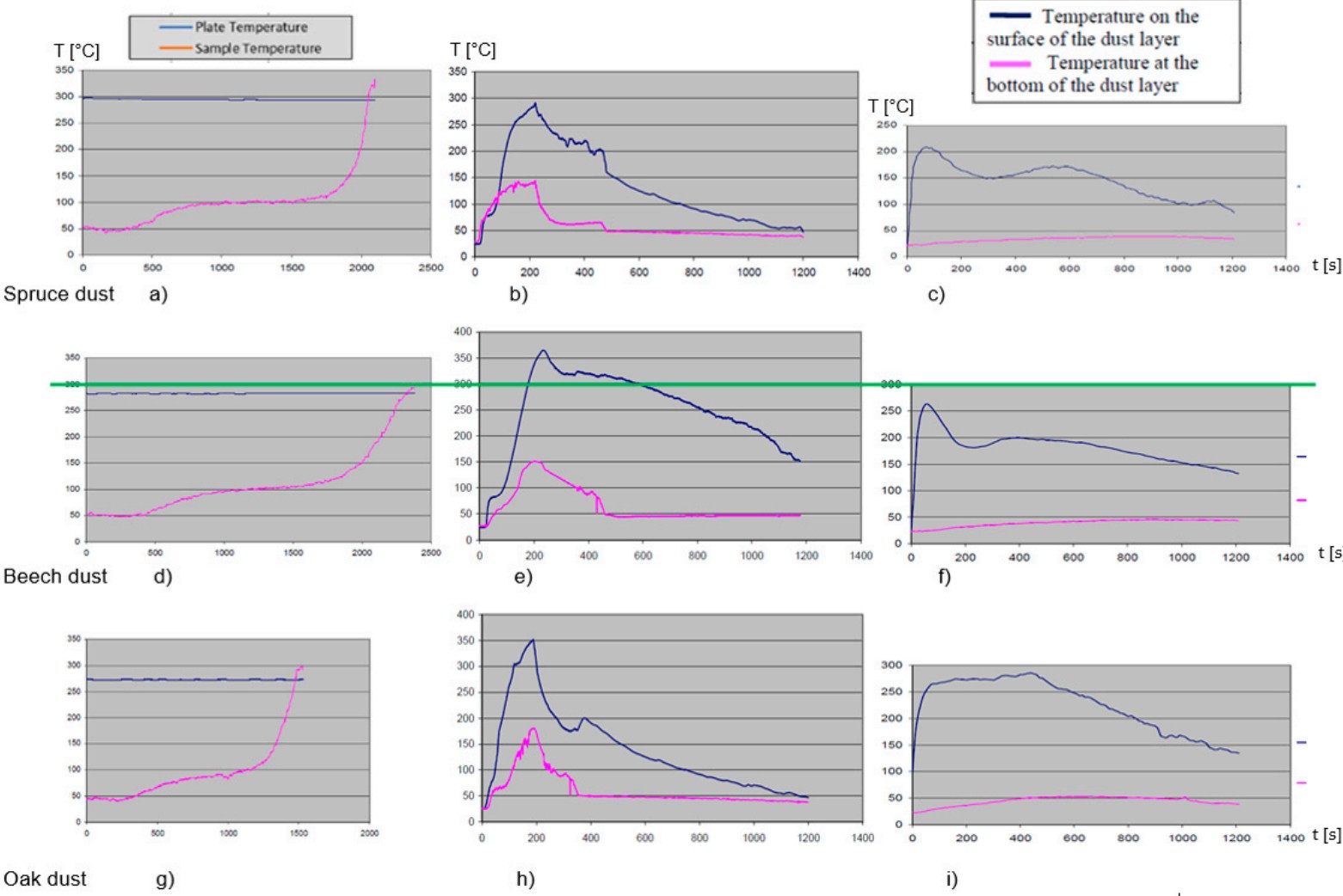

**Figure 6.** Test data showing an ignition for the samples. Legend: (**a**,**d**,**g**) hot plate source, (**b**,**e**,**h**) electric coil, (**c**,**f**,**i**) cigarette.

The ignition behaviour of the individual waste wood dust samples was very similar. Differences were in the minimum ignition temperature values and in the time intervals between the observed ignition phases. When the dust layer temperature reached about 100 °C, smouldering nests began to form in each sample. With subsequent temperature increase, thermal dust degradation spread along the perimeter of the deposited layer and towards the centre of the sample (Figure 6). Subsequently, the increasing temperature of waste wood dust was recorded on thermocouple No. X2, until the temperature of the hot surface was exceeded (Figure 6a,d,g).

Table 3 shows the measured values of minimum ignition temperatures and the behaviour of the layer during the ignition of waste wood dust.

**Table 3.** Hot surface temperature results.

| Dust Samples | Minimum Ignition Temperature (°C) | Visual Observations During Measurements |
| --- | --- | --- |
| | | Observations or Comments |
| Spruce | 300 | – 19 min 50 s, 2 smouldering nests were formed, spreading from the edges towards the centre of the sample, fuming<br>– 22 min 10 s, heating observed, white ash falling off<br>– 34 min 20 s, sample is completely charred |
| Beech | 290 | – 23 min 35 s, 3 smouldering nests were formed, spreading from the edges towards the centre of the sample, charring, fuming<br>– 26 min 50 s, heating around the perimeter of the sample,<br>– 37 min 45 s, sample is completely charred |
| Oak | 280 | – 16 min 20 s, 2 smouldering nests were formed, significant fuming<br>– 19 min, heating around the perimeter of the sample,<br>– 25 min, sample is completely charred |

A difference occurs in the time of realisation of these events. Spruce dust reached the hot plate surface temperature in 33 min (Figure 6a) and the 50 °C difference was not exceeded. Hehar [64] confirmed the results. The fine dust fraction (dust particles passing through 90 μm sieve) had the higher ignition risk compared to the medium (between 90 and 180 μm) and coarse (between 180 and 420 μm) dust fractions [64]. Hehar et al. [64] investigated the ignition risk of dust from loblolly pine wood by quantifying (including moisture content and grinding screen size effects) the amount of dust in ground wood chips, and the hot surface ignition temperature.

In the case of beech waste wood dust, the process was slower. The surface temperature was reached in 37 min (Figure 6d). The surface temperature of oak dust was reached in 25 min (Figure 6g) at 208 °C. The given minimum ignition temperature of oak is lower due to the particle composition of oak dust [12], where more than 56% of the volume of particles is 0.71 μm.

Beljaková et al. [65] recorded a minimum ignition temperature of beech dust of 330 °C in a 5 mm layer. Determination of the minimum ignition temperature of dust in the deposited state is described in detail in their paper. The measurements were carried out with beech dust obtained in various technologies (belt sander and GBS 100 AE). Differences in the ignition temperatures of these samples were not confirmed (Beliaková et al., 2009) [65]. Determination of the minimum ignition temperature of dust layer was very clearly described and explained by Hosseinzadeh et al. [66], Fernandez-Anez et al. [67] and Danzi et al. [68].

Selected ignition sources cause thermal degradation of dust samples. Contact of solid hot surfaces with flammable substances presents a risk of fire or explosion due to the spontaneous ignition of the formed explosive/combustible mixture [69]. The ability of a heated surface to cause ignition depends on the type and concentration of the mixture of flammable substance with air. This ability also increases with the increasing temperature and surface area [70,71]. In addition to easily recognisable hot surfaces such as radiators, drying ovens, heating coils and others, hazardous temperatures can also occur during mechanical and machine processing. These processes include equipment, protective systems and components that convert mechanical energy into heat [72,73].

### 3.3.2. Monitoring of Waste Wood Dust Ignition by Electric Coil

In the case of ignition by an electric coil, the temperature course was very similar in all wood dust samples (Figure 6b,e,h). After 30 s ignition, the layer of dust was charred at the points between the arms of the inserted electric coil, accompanied by heavy smoke. A glowing nest was formed around the hot wire and over time, the charred layer spread further over the surface of the dust layer.

The thermal dependence is seen in the interruption of the electric current through the coil at 180 s when both temperatures reached their maximum values. The temperature on the surface of the dust layer exceeded 300 °C while on the bottom of the layer it was 150 °C.

Once the initiation process was stopped, the behaviour of dust samples was observed for another 20 min. The charring process was comparable in all dust samples, but there was a significant difference in the size of the charred area (Figure 7a–c).

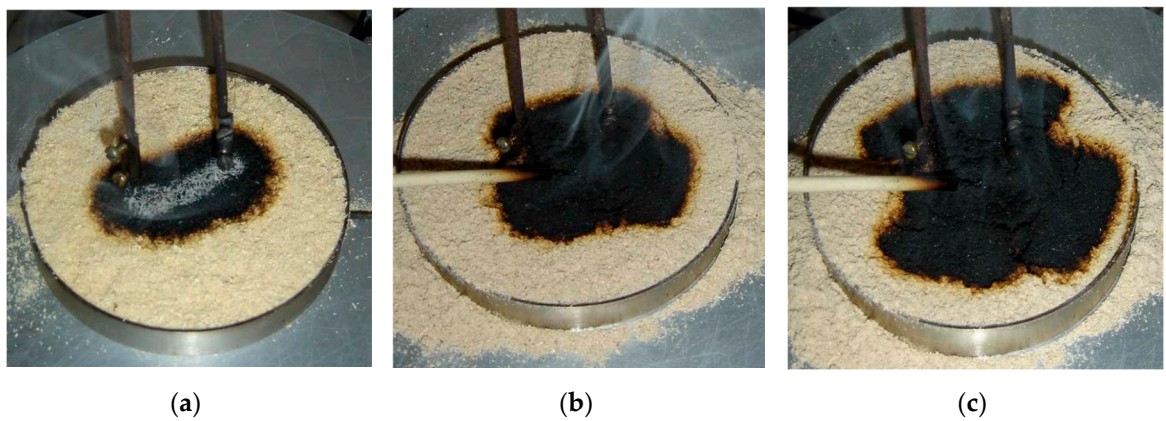

(**a**)　　　　　　　　　　　　　　　　　　(**b**)　　　　　　　　　　　　　　　　　　(**c**)

**Figure 7.** Charred surface of dust ignited by an electric coil, documented by camera (Olympus PEN E-PL8 + ED 14-42 II R lens, Olympus Czech Group, s.r.o., Czech Republic). Legend: (**a**) spruce dust, (**b**) beech dust, (**c**) oak dust at 3rd min.

The oak dust had the largest charred area as well as the largest glowing area around the coil. In addition, the surface temperature of the oak dust layer did not drop below 200 °C even after 20 min which was caused by the strong heating of the entire layer of tested dust surface.

In the case of ignition by an electric coil, the samples were subjected to heating at the local hot spot of the heated spiral wire. The samples burned into depth, suggesting that a hot wire is a great danger, especially if it is covered with a layer of dust. In particular, thinner layers of deposited dust on electric conductors pose a great danger [74,75].

### 3.3.3. Monitoring of Waste Wood Dust Ignition by a Smouldering Cigarette

Smouldering is a heterogeneous combustion where oxygen molecules (a gaseous oxidant) affect the surface of a combustible solid substance—tobacco. The combustion reaction takes place on the surface of the combustible substance [76].

Two thermocouples, as in previous experiments, were used to monitor the surface and bottom temperature of the dust layer. The depicted thermal dependence of wood dust layers in the case of ignition by a smouldering cigarette, shows a significant difference in the oak sample (Figure 6c,f,i). [71].

The smouldering cigarette only affected the surface of the dust (Figure 8), the fire did not burn through the sample layer and the temperature at the bottom of the dust layer increased only slightly. Upon application of the cigarette, a charred layer of dust was formed at the tip of the burning cigarette and it progressed in the direction of the burning cigarette. The figures show that the surface temperature increased, although the smouldering part did not act at a given location for about 100 s. After the cigarette had completely burned out, the charred layer gradually expanded, with strong smoke and smouldering observed in oak dust.

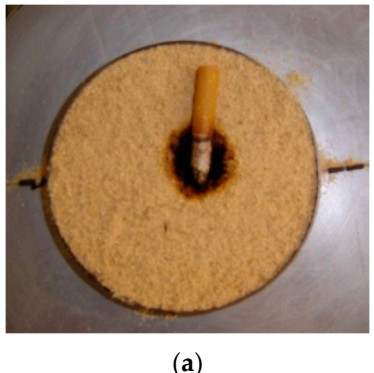 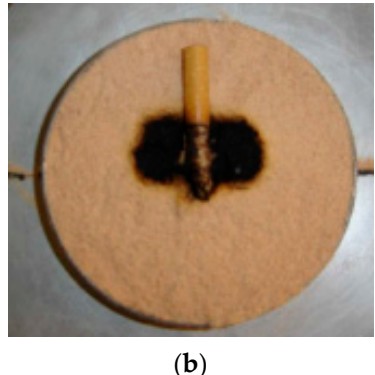 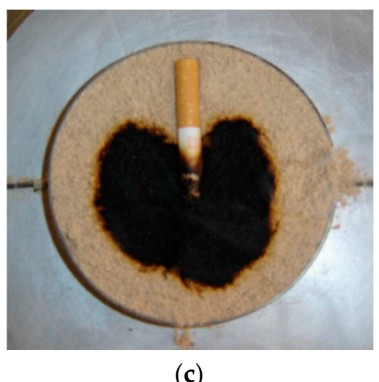

(**a**) (**b**) (**c**)

**Figure 8.** The charred surface of dust ignited by a smouldering cigarette, documented by a camera (Olympus PEN E-PL8 + ED 14–42 II R lens, Olympus Czech Group, s.r.o., Czech Republic). Legend: (**a**) spruce dust, (**b**) beech dust, (**c**) oak dust at the 3rd min.

The layer of spruce dust was charred (Figure 8a) only at the area of application of the smouldering tip of the cigarette, followed by light smouldering, without the tendency to expand the charred area. When the beech dust sample was ignited (Figure 8b), the area of the charred layer was larger and the smouldering more intense. At the ignition of the oak dust sample (Figure 8c), the charred surface rapidly spread across the surface with multiple smouldering locations, and the sample burned all the way through, which was also indicated by an increase in temperature at the bottom of the dust layer.

The charred areas ignited by a smouldering cigarette (Figure 8 and Table 4) were significantly smaller than those ignited by the electrical coil (Figure 7 and Table 4). A smouldering cigarette poses a fair ignition risk especially for fine dust mixtures (Figure 8c).

**Table 4.** Comparison of the influence of ignition sources on selected dust samples.

| | **Ignition Source** | **Spruce Dust** | **Beech Dust** | **Oak Dust** |
|---|---|---|---|---|
| Hot surface | Minimum ignition temperature (°C) | 301.6 ± 2.870 | 290.8 ± 2.227 | 280 ± 0.2.280 |
| | Ignition time (s) * | 2060 | 2262 | 1500 |
| Electric coil | Charred area (%) | 18.02 ± 0.487 | 40.01 ± 0.303 | 70.02 ± 0.337 |
| | Ignition time (s) * | 180 | 180 | 180 |
| Smouldering cigarette | Charred area (%) | 5.04 ± 0.102 | 40.01 ± 0.303 | 40.06 ± 0.241 |
| | Ignition time (s) * | 1200 | 1200 | 1200 |

* ignition time refers to 100% of dust layer area being charred.

Summarising the results is difficult. The experiments were carried out on the same hot plate apparatus with the addition of electric coil and smouldering cigarette as ignition sources. Hot plate apparatus is recognised by the academic community as suitable equipment for testing the minimum ignition temperature of a deposited layer of dust [47,51]. The first experiments analysed the behaviour of the dust layer on the surface of the hot plate. The obtained results are in alignment with the research of other authors [47]. The difference in temperature curves (Figure 7a,d,g) is related to the diversity of wood dust (spruce, oak, beech) and the diversity of the granulometric composition of wood dust (prepared by sawing and grinding).

Marková et al. [20] analysed the granulometric composition of various types of wood dust produced by a circular saw and statistically evaluated the influence of wood species and particle size on the granulometric composition of wood dust. Differences in the dust structure of individual tree species were not shown to be statistically significant. Barcík et al. [77] confirm this finding. Barcík [77] also notes that the species of wood has not shown a clear effect on the course of the granulometric composition of the wood when working with different tools.

Differences in the dust structure of individual tree species are shown to be statistically significant among the fractions [45]. This fact is also confirmed in the research of Saejiw et al. [78]. The results show the influence of the granulometric composition of individual samples on the course of thermal



degradation in all ignition sources. The temperature curves obtained via hot surface apparatus show slower thermal degradation of waste wood dust and the spruce and beech samples. Spruce waste wood dust, which also contains sawdust (95%) of particles with a size of more than 200 μm), underwent thermal degradation in 2060 s (Table 4). The slowest degradation occurred in beech dust, with the dominating particle segment (98%) above 090 μm. Fine spruce dust reached the hot surface temperature in half the time (1500 s).

The effect of the particle composition can also be seen in thermal degradation caused by an electric coil (Figure 6b,e,h and Figure 7a–c). The most noticeable difference is seen in the ignition by a smouldering cigarette (Figure 8). Based on the given area, it is possible to identify a linear dependence in the increase of the area of wood dust under the same thermal conditions.

## 4. Conclusions

In general, the oak dust sample exhibited the worst items, which are the lowest the minimum ignition temperature, larger charred areas, in the shortest time required for its ignition.

Experiments confirmed the assumption based on the results obtained by the sieve analysis and determination of moisture content of samples: the sample with the finest granulometric composition and the lowest humidity is the most combustible.

The most effective ignition source was an electrical coil which due to the spatial arrangement provided sufficient amounts of ignition energy to the samples and was the only ignition source that resulted in the dust sample burning all the way through the layer.

All ignition sources which were used lead to in the charring of the sample, varying in the size of the charred area, in this order: electric coil, smouldering cigarette and hot surface.

Deposited dust may burn in a form of flaming combustion, glowing or smouldering. All of the above forms were observed with an ounce of ignition by an electric coil.

**Author Contributions:** Conceptualization, I.T. and I.M.; methodology, investigation and resources I.T.; writing—original draft preparation, I.M.; writing—review and editing I.T. and I.M. All authors have read and agreed to the published version of the manuscript.

**Funding:** This paper was supported by the Cultural and Educational Grant Agency of the Ministry of Education, Science, Research and Sport of the Slovak Republic on the basis of the project KEGA 0014UKF-4/2020 Innovative learning e-modules for safety in dual education.

**Acknowledgments:** This article was supported by the Project KEGA 0014UKF-4/2020 Innovative learning e-modules for safety in dual education.

**Conflicts of Interest:** The founding sponsors had no role in the design of the study, in the collection, analyses, or interpretation of data; in the writing of the manuscript and in the decision to publish the results.

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
