# Peer review of "Ignition of Deposited Wood Dust Layer by Selected Sources"

_applsci, doi:10.3390/app10175779_

Round 1

Reviewer 1 Report

25-26                  minimum ignition temeperatures: what is the significance of these numbers? Is there any statistics?

26                        What is meant by ignition mechanism?

108                      Is there any rationale fort he value of 2/3 for the safety coefficient?

111-112              What’s the difference between explosion and detonation?

114-118              combination of 3 wood types, 3 ignition sources, contact area, and duration of contact suggest to provide some statistics. This is not clear in the rest oft he paper.

120-125              This belongs to the introduction

136                      will be described o rare described in [43]?

147/48 and 152                            Repeated naming of sieve apparatus

158/159             name of hot plate apparatus

163                      Details of the hot plate apparatus: arrangement of temperature measurement are missing

165/166             This is not clear: minimum ignition energy = temperature? Anticipating outcome of the experiment with the design?

180                      Melting?

181                      How can it cool down, when it seems to be constantly heated?

183                      Why 15 mm?

206                      Here a repetition of measurements is stated. Is this the same fort he other settings? Does this produce some statistics?

219/220             not have asignificant effect: Was this verified? Reference?

272                      Probably Figure 7.

382                      Fig. 9, not 8c?

Author Response

Thank you for the helpful comments we have incorporated into the article. We believe that editing the article has made the article better. The article has major adjustments because we have taken into account the comments of other reviewers. One reviewer requested to shorten the introduction and he wanted the change another parts from results to methods.

Our new comments in article are highlight of yellow colour.

Our comments:

Line: 25-26 Yes, we added in Abstract: as an important property of any fuel because the combustion reaction of the fuel becomes self-sustaining only above this temperature

Line 26:  Sorry, we changed mechanism for process

Line108 This information is published of Damec, J. Anti-explosion prevention. 1st ed., ], EDICE SPBI SPEKTRUM, Ostrava, Czech Republic, 1998, 188 p. (in Czech)

Line 111-112: We added this difference in article.

Line 114-118: research follows up on solved Slovak research projects of wood dust.

Line 120-125: Yes, we agree and we realised this change.

Line 136: We added authors.

Line 147/48 and 152:  Thanks. We repaired it

Line 158/159: We added it.

Line 163: We added it.

Line 165/166:We repaired it

Line180: We correct it.

Line 181: It was mistake, We correct it.

Line183: We added it

Line 206: In this step of research - no

Line 219/220: We added it.

Yes, it was verified by Pastier, M., Tureková, I., Turňová, Z., & Harangozó, J. (2013). Minimum Ignition Temperature of Wood Dust Layers. Research Papers Faculty of Materials Science and Technology Slovak University of Technology, 21(Special-Issue), 127-131.

Line 272: It was mistake. Thanks

Line382: It was mistake. Thanks

Reviewer 2 Report

Dear Authors,

The masnuscript is in my opinion to long and have to be shortened by reorganization of the Introduction chapter and appropriate presentation of results. You should also you should opt out of three-level division of the text into chapters. It only gives an apparent order of text. It should be clearly ordered without such a division.

Speciffic comments:

L.20 Why don't you use the name Picea abies L.?

L. 24 Sieve analysis can be done but as a result of it the particle size distribution can be determined.

L. 29 Th abstract have to be related to the keywords and they have to mentioned in Abstract? What about granulometry and hot plate assembly?

L. 115 Detete the word samles and desribe the samples in chapter Method

L. 115-118 Methodical assumtions.

L.120-125 Has the forestry of Slovakia any meaning in the methods of determination of of wood dust ignition? Please remove this part of the text or put it into Introduction.

L.133 Errors in the Polish name of manufacturer.

L. 137-139 Bosch or BOSCH? Specify the name and place of manufacturer.

L. 148 the standard cited (45) seems to be not appropriate for sieve analisys. Consider the standard ISO 2591-1:1998

L. 148-149 How about the vaccum cleaner and its bag used in the experiments?

L. 152-157 Description of sieve analysis. Compare L. 147-148.

L. 158-159 How about a machine, manufacturer and country?

The subchapter 2.3 should be moved to L. 158 I think.

L. 165 Dust used in the study was not waste. It was produced specially in this purpose. (L. 132, 137, 139).

Fig. 1. Pictures of dust samples are not relevant for dust charcterization. Bulk density or tapped bul density can be better in this puropse.

L. 189 (and others) Please use SI symbols of units (s).

Subchapter 2.4 please unify the symbols of measurement points X1 X2 1 2.

L.202 Is this subchapter really neccesary?

Subchapters 3.1, 3.2, 3.3 It is already known that they relate to the results.

L. 211-218 This part of text describes mathods, not results.

L.218 Higroscopic properties of wood dust in relation to its size were already studied https://doi.org/10.1016/j.powtec.2019.02.007

Subchapter 3.2 The results of the sieve analysis should be presented in one chart by three curves. The way of presentation of these results is inappropriate. I strongly suggest to present them according to the standard ISO 9276-1.

L. 232-250 Results of the dust sieve analysis should be discussed only in the context of dust ignition.

L.252-253 Method description, not results. Has the picture of microstructure any influcence on dust ignition? If not, please don't take it into account.

L.279-291 Methodology again. Please  remove all methodical descriptions from the results chapter.

Fig. 7 Check the chart e.

L.330-333 Method again.

L.355-360 Is this part of the text really neccesary?

Author Response

Thank you for the helpful comments we have incorporated into the article. We believe that editing the article has made the article better. we have taken all the comments into account, shortening the introduction. Our new comments in article are highlight of green colour.

Our comments:

L.20 Sorry, we change this therm.

L. 24 Yes, we added it in Abstract.

L. 29 Sorry, we repared it.

L. 115 We added it.

L. 115-118 We repaired it.

L.120-125 We repared it. It is the same requirement as had another reviewer.

L.133 We repared it.

L. 137-139 We added it.

L. 148 We repared it.

L. 148-149 We added it.

L. 152-157 We repared it.

L. 158-159? We added it.

The subchapter 2.3 should be moved to L. 158 I Yes, We repared it.

L. 165 Dust used in the study was not waste. It was produced specially in this purpose. We added this information.

Fig. 1. Pictures of dust samples are not relevant for dust charcterization. Bulk density or tapped bul density can be better in this puropse.

            Sorry, we did not realise bulk density.

L. 189 Thanks.

Subchapter 2.4 please unify the symbols of measurement points X1 X2 1 2. yes, we did it.

L.202we think so, because the above modification of the method is our invention

Subchapters 3.1, 3.2, 3.3 It is already known that they relate to the results. We repaired it.

L. 211-218 We repaired it.

L.218 We added it.

Subchapter 3.2

Thank you for your suggestion. Fig. 4 remained in the article because the other reviewers had not reservations but we added a presentation according to the standard ISO 9276-1.

L. 232-250 We accepted it.

L.252-253 We accepted it.

L.279-291 We repaired it.

Fig. 7 Thanks, it is correct.

L.330-333 We repaired it.

L.355-360 We repaired it.

Reviewer 3 Report

The paper does well investigated the effectiveness of different ignition sources to ignite different wood dust samples. The work approach provide a clear view of the potential hazards of the ignition sources, taking into account the different characterization of the dust samples; do offer a comprehensive analysis of the dust generation capability of different tools, commonly used in the wood industry.

Nevertheless the content and the research design could be intensively improved.

For all the revisions see the attached manuscript.

A major work of literature search could be done to improve the reference of the work with respect to the dust explosion science. A list of possible major references in this field is reported below.

Moreover, the experimental tests (those which are not standardized) could be described better and more in details, as to make them reproducible for sake of comparison with future works. As an example the charred-sigarette test could be described more in details (how is the charred area measured?)

The different ignition sources could be identified by their energy release (or energy density release), could the authors add these data?

The reviewer suggest to stress more these "non standard" experimental conditions, which are interesting and promising, trying to define some "key" parameters to be used as reference (could it be the dT/dt max or the Tmax, or the time to these two values?)

It is important to verify the main literature in dust explosion science field also as to report the dust explosion fundamental concepts better inside the work (see underlined expressions in the scanned copy).

Equation 1 is derived from where? (need citation)

Some references (just as an example):

Amyotte, P. R. An Introduction to Dust Explosions (2013). Elsevier

Hosseinzadeh, S., Norman, F., Verplaetsen, F., Berghmans, J., & Van den Bulck, E. (2015). Minimum ignition energy of mixtures of combustible dusts. Journal of Loss Prevention in the Process Industries, 36, 92–97. https://doi.org/10.1016/j.jlp.2015.05.012

Fernandez-Anez, N., Slatter, D. J. F., Saeed, M. A., Phylaktou, H. N., Andrews, G. E., & Garcia-Torrent, J. (2018). Ignition sensitivity of solid fuel mixtures. Fuel, 223, 451–461. https://doi.org/10.1016/J.FUEL.2018.02.106

Ennis, T. (2016). Fire and explosion hazards in the biomass industries. Institution of Chemical Engineers Symposium Series, 2016 (161), 1–9.

Minimum Ignition Temperature of layer and cloud dust mixtures. E Danzi, L Marmo, D Riccio. Journal of Loss Prevention in the Process Industries 36, 326-334

Author Response

Thank you for the helpful comments we have incorporated into the article. We believe that editing the article has made the article better. We have taken all the comments into account, shortening the introduction. Our new comments in article are highlight of blue colour. First reviewer has yellow and second reviewer green colour.

We tried to repaired all underlined expressions in the scanned copy.

This report came third. In fact, corrections have already been made according to what the two opponents request. The introduction was reduced, the methodology completed, the results of dust initiation presented in the context of sieve analysis.

Thank you for the term of "Dustiness".

Thanks for offer very interesting references.

Round 2

Reviewer 2 Report

Dear Authors,

You put a lot of effort into improvement of the manuscript. I appreciate it very much. The manuscript sounds now much better.

But I'm still thinking that fig. 4. shuold be removed because: it presents the same data as fig 5.; the fig 5 is much clearer, a reader can rapidly evaluate which dust is finer and which is coarser; the charts in fig 4 are inconsistent with the standard - where is the 0 point of X-axis???; data presentation as in fig 5 takes up less space.

There are new citations added. But some of them seem to be inaccurate. For example the new citation 13 concerns the fire and explosion hazard, but not the harmful effects on the human organism as in the phrase. Please check carefully the correctness of the citations and provide appropriate ones.

Author Response

Dear Reviewer,

Thank you for your comment.

We accept the above request and fig. 4 has been removed. The second reviewer alerted us to the unfinished course of cumulative curves. The curves were completed and added with an inlet sieve.

Citation 13 was corrected.

Reviewer 3 Report

All the suggestions/indications by the reviewer have been incorporated and almost completely fixed.

Cumulative curve of Figure 5 are not correct (all must go from 0 to 100%)

Some references are not reported fully corrected (Standard needs the complete title (e.g. EN 50281-2-1:1999...)

The reviewer suggest a good review of English throughout all the manuscript.

Author Response

Dear Reviewer,

Thank you for your comment.

Thanks for the alert in Figure 5. The cumulative curve was adjusted and added an inlet sieve. The inlet sieve was 1 mm in size. A whole sample of all the dust passed through this sieve.

At the repeated request of the second Reviewer, Figure 4 was removed and only the cumulative curve was left.

We corrected EN 50281-2-1:1999. We apologize for the mistake. The year 2002 was written, because EN 50281-2-1 entered into force in the Slovak Republic in 2002.